# Real-World Experience with Pasireotide-LAR in Cushing’s Disease: Single-Center 12-Month Observational Study

**DOI:** 10.3390/jcm14082794

**Published:** 2025-04-18

**Authors:** Lukasz Dzialach, Wioleta Respondek, Przemyslaw Witek

**Affiliations:** 1Department of Internal Medicine, Endocrinology and Diabetes, Medical University of Warsaw, 03-242 Warsaw, Poland; 2Department of Internal Medicine, Endocrinology and Diabetes, Mazovian Brodnowski Hospital, 03-242 Warsaw, Poland

**Keywords:** cortisol, Cushing’s disease, hyperglycemia, pasireotide, pituitary tumor

## Abstract

**Background/Objectives**: Pasireotide-LAR represents a novel therapeutic option for patients with Cushing’s disease (CD). Its efficacy and safety were assessed in clinical trials; however, the real-world evidence is still scarce. **Methods**: The study aimed to evaluate the impact of 12-month pasireotide-LAR therapy on disease control, glucose metabolism, lipid profiles, and adverse effects in a real-life setting. We retrospectively studied prospectively collected data of patients with persistent or recurrent CD administered with pasireotide-LAR in a single pituitary center. **Results:** Mean urinary free cortisol (mUFC) showed a sustained decrease from baseline, with the most pronounced decrease in the first 3 months of therapy (*p* = 0.007). The analysis of mean late-night salivary cortisol showed fluctuations over time, with the largest mean reduction in mLNSC at 3 months. During the therapy, an improvement in blood pressure control was observed, with a significant decrease in systolic blood pressure during the first 6 months of treatment (*p* = 0.005). Hyperglycemia was the most common adverse effect. Fasting plasma glucose and glycated hemoglobin (HbA1c) showed a gradual increase during pasireotide-LAR treatment, with the HbA1c significantly increasing at the last follow-up (*p* = 0.04). **Conclusions**: Pasireotide-LAR is an effective alternative treatment in selected patients with CD. Pasireotide-LAR is overall safe and well tolerated, with hyperglycemia being the most common but manageable adverse event.

## 1. Introduction

Adrenocorticotrophin (ACTH)-secreting pituitary tumors are the leading cause of endogenous Cushing’s syndrome (CS) and traditionally are defined as Cushing’s disease (CD) [1,2]. Both in the active phase of the disease and after achieving remission, CD is associated with the presence of various comorbidities and a significantly reduced quality of life [3,4]. Optimization of the long-term outcomes in CD requires an early accurate diagnosis, careful selection of treatment, and management of the disease and its comorbidities by experienced clinicians [1,4].

Transsphenoidal selective surgery (TSS) is considered a first-line treatment, with a remission rate of approximately 80% in patients with micro- and 60% with macroadenomas, if the procedure is performed by an experienced neurosurgeon [1]. For patients with non-radical surgery or recurrent disease and when surgery is contraindicated, not feasible, or not accepted, in recent years, the possibility of pharmacotherapy has emerged as a second-line treatment due to the availability of new drugs. The medical agents utilized in CD treatment target adrenal steroidogenesis, somatostatin and dopamine receptors in the pituitary gland, and glucocorticoid receptors [5,6].

Pasireotide is a second-generation somatostatin receptor ligand (SRL) with the highest affinity for somatostatin receptor subtype 5 (SSTR5), the most abundantly expressed SSTR in corticotropinomas [7,8,9]. Pasireotide long-acting release (LAR) is a modified, long-acting version of pasireotide which is administered intramuscularly once monthly, with a similar efficacy and safety profile as the short-acting form [5,8]. In a phase 3 clinical trial (NCT01374906), at month seven, mean urinary free cortisol (mUFC) normalization was achieved in 41% of CD patients [8].

The data from clinical practice on patients with CD treated with pasireotide are limited, and most of them concern the subcutaneous formulation of pasireotide [10,11,12,13], with isolated data on pasireotide LAR [14]. Hence, in this study, we aim to present the impact of 12 months of pasireotide LAR treatment on the disease control, cardiovascular parameters, and glucose and lipid profiles in patients with persistent or recurrent CD at a single center in Poland in a real-world setting.

## 2. Materials and Methods

### 2.1. Study Population

We identified 8 patients with active CD treated with pasireotide LAR under the National Health Fund drug program in the Department of Internal Medicine, Endocrinology, and Diabetes of the Medical University of Warsaw. The patients were evaluated between January 2022 and June 2024 during routine outpatient visits by the drug program schedule. Six patients received pasireotide LAR for at least 12 months and were considered suitable for the analysis. The patients’ detailed characteristics before and during the pasireotide LAR treatment are summarized in Table 1.

### 2.2. Study Protocol

A real-world retrospective study was conducted at a single endocrine center in Poland (Department of Internal Medicine, Endocrinology, and Diabetes of the Medical University of Warsaw). The data were prospectively collected between January 2022 and June 2024. Patient visits occurred monthly and were focused on clinical assessment, including CD symptoms, glucose metabolism evaluation, and possible side effects of pasireotide-LAR treatment. Complete biochemical and hormonal measurements were conducted at baseline, during months 3, 6, 9, and 12, and at the last follow-up (LFU). The hormonal evaluation included measurements of ACTH, morning and midnight serum cortisol, urinary free cortisol (UFC), and late-night salivary cortisol (LNSC). The mean UFC (mUFC) was calculated from two or three 24 h UFC measurements, and the mean LNSC (mLNSC) from two LNSC measurements which were collected on three or two consecutive days before the next pasireotide-LAR dose. The serum cortisol (normal range: morning, 3.7–19.7 μg/dL; midnight, <5.4 μg/dL) and UFC (normal range: 4.3–176.0) were measured with the Abbott Alinity immunoassay (Chicago, IL, USA). The plasma ACTH (normal range: 4.7–48.8 pg/mL) was measured with the LIAISON XL chemiluminescent immunoassay (Saluggia, Italy). The LNSC (normal range: <2.5 ng/mL) was measured using ELISA with the MINDRAY ELISA READER-96A (Shenzhen, China). The initial dose of pasireotide LAR was 10 mg intramuscularly every 28 days. The pasireotide LAR was up-titrated (to max 40 mg every 28 days) if the mUFC and/or mLNSC persisted above the upper limit of normal (ULN). Biochemical control was defined as the normalization of either mUFC or mLNSC.

The assessment of glucose metabolism included the patients’ blood glucose self-monitoring every week, fasting plasma glucose (FPG) monthly, and glycated hemoglobin (HbA1c) level every 3 months. The investigators were permitted to initiate or adjust antidiabetic medications to manage hyperglycemia during the study. Additionally, the liver function parameters of alanine transaminase (ALT), aspartate transaminase (AST), gamma-glutamyltransferase (GGT), and bilirubin levels, as well as the lipid profiles, were measured every 3 months.

The aim of abdominal ultrasound was primarily to detect cholelithiasis. It was performed at baseline and after 6 and 12 months of treatment. Pituitary magnetic resonance imaging (MRI) was performed at baseline and after 12 months of treatment. An electrocardiogram (ECG) with a calculated corrected QT (QTc) interval was performed at baseline and monthly for the first three months following the start of treatment, and then every three months.

### 2.3. Ethical Considerations

This retrospective study was conducted in accordance with the Declaration of Helsinki. Written informed consent was obtained from all patients for the use of the clinical data in the study after providing all information regarding the purpose of the study. The investigators adhered to good clinical practice guidelines. The study was approved by the Ethics Committee of the Medical University of Warsaw (permission number: AKBE/335/2024).

### 2.4. Statistical Analysis

Statistical analysis was carried out using the Prism 9.0 software package (GraphPad Software, Inc., Bonstron, MA, USA). The differences between the timing points were analyzed with a one-way ANOVA, followed by a Dunnett’s post hoc test. The level of statistical significance was set at a *p*-value < 0.05. The results were expressed as means and SDs.

## 3. Results

### 3.1. Characteristics of Study Population

Six patients (five women) received pasireotide LAR for at least 12 months (the mean [SD] pasireotide LAR exposure was 16.6 [6.40] months) and were considered suitable for the analysis. The study cohort’s mean (SD) age was 45.8 (9.6) years. All patients but one harbored a pituitary microadenoma (in three patients, the pituitary tumor was not detectable on MRI). At diagnosis, all patients underwent TSS as a first-line treatment; three patients had persistent and three patients had recurrent disease. Two patients underwent repeated TSS, and three received previous pituitary radiotherapy. Three patients were previously treated pharmacologically (including osilodrostat, ketoconazole, and temozolomide). The mean (SD) mUFC and mLNSC at study entry were 185.68 ug/24 h (41.11) and 9.90 nmol/L (4.10), respectively. The detailed patient characteristics are summarized in Table 1.

### 3.2. Effectiveness and Hormonal Parameter Analysis

The analysis of mean (SD) mUFC showed a significant decrease from a baseline of 185.68 ug/24 h (43.11) to the subsequent time points, with the last observed value (LOV) showing a mean (SD) mUFC of 91.93 μg/24 h (56.54). The post hoc Dunn’s test revealed statistically significant reductions in the mUFC levels at all analyzed time points. The analysis of mean mLNSC levels showed fluctuations over time, with a baseline mean (SD) of 9.90 nmol/L (4.10) and an LFU of 6.85 nmol/L (5.04); the analysis of differences relative to the baseline revealed the largest mean reduction in mLNSC at 3 months.

The analysis of morning serum cortisol showed stable mean (SD) values over time, ranging from a baseline of 14.62 μg/dL (4.24) to 13.13 μg/dL (4.67) at the LFU. The analysis of mean (SD) midnight serum cortisol showed a decrease from a baseline of 16.22 μg/dL (4.64) to 9.75 μg/dL (3.77) at the LFU. The analysis of morning ACTH levels showed relatively stable mean (SD) values over time, ranging from a baseline of 48.38 pg/mL (11.17) to 53.51 pg/mL (30.68) at the LFU. The post hoc Dunn’s test comparing the baseline levels of mLNSC, morning and midnight serum cortisol, and ACTH to the subsequent time points revealed no statistically significant differences.

A complete hormonal evaluation at each time point during pasireotide LAR treatment is detailed in Table 2 and Figure 1.

### 3.3. Effects on Metabolic and Cardiovascular Parameters

The analysis of body weight and BMI revealed a gradual decrease in the mean [SD] values from the baseline (81.70 kg [13.08] and 29.33 kg/m^2^ [5.80], respectively) to the LFU (77.30 kg [8.83] and 27.91 kg/m^2^ [4.00], respectively). However, the post hoc Dunn’s test comparing the baseline to the subsequent time points indicated no statistically significant differences. The analysis of SBP and DBP showed a reduction in the mean [SD] values from the baseline (146.00 mmHg [13.04] and 95.83 mmHg [13.06], respectively) to the LFU (126.67 mmHg [9.00], 89.67 mmHg [9.16], respectively). The post hoc analysis revealed statistically significant differences for the SBP at 6 months, 9 months, and LFU, but no significance was observed for the DBP. A complete evaluation of the metabolic and cardiovascular parameters at each time point during pasireotide LAR treatment is detailed in Table 3 and Figure 2.

### 3.4. Effect on Lipid Profiles

The analysis of TC levels showed a general decrease in the mean [SD] values from the baseline (222.50 mg/dL [61.48]) to the LFU (156.83 mg/dL [38.35]), with some fluctuations observed at intermediate time points. A statistically significant decrease was observed at the LFU. The analysis of LDL-C and HDL-C levels showed a decrease in the mean [SD] values from the baseline (134.6 mg/dL [53.74] and 50.47 mg/dL [9.67], respectively) to the LFU (87.50 mg/dL [33.41] and 42.67 mg/dL [12.71], respectively), with some fluctuations observed at intermediate time points. A statistically significant decrease was observed for the HDL-C at 9 and 12 months. The analysis of TG levels showed a gradual decrease in the mean [SD] values from the baseline (185.82 mg/dL [62.44]) to the LFU (128.50 mg/dL [45.76]); however, there was no statistical significance at subsequent time points. A complete evaluation of the lipid profiles at each time point during pasireotide LAR treatment is detailed in Table 3.

### 3.5. Effect on Glucose Metabolism

The baseline mean (SD) FPG level was 93.90 mg/dL [10.65], and the mean (SD) HbA1c was 5.59% [0.33]. In the analyzed group, initially, five patients presented with glucose metabolism impairment (two patients had diabetes, and three had prediabetes) and were receiving hypoglycemic medications (three patients were taking metformin, and two patients were taking semaglutide). The analysis of FPG and HbA1c showed a gradual increase in the mean [SD] values from the baseline to the LFU (113.67 mg/dL [27.33] and 6.57% [0.89], respectively). The most considerable FPG level and HbA1c increases from the baseline was observed at the LFU. A statistically significant increase in HbA1c was observed at LFU; however, the changes in the FPG level did not reach statistical significance. These results indicate a general upward trend in the FPG level and the HbA1c, with more pronounced increases at later time points. A complete evaluation of the glucose metabolism parameters at each time point during pasireotide LAR treatment is detailed in Table 3 and Figure 3.

In all patients, hyperglycemia-related adverse events (AEs) occurred; one initially prediabetic patient was diagnosed with diabetes. The remaining patient who had a normal glucose metabolism at study entry developed impaired fasting glucose during pasireotide-LAR treatment, which was managed with diet therapy. None of the patients in the current study dropped out due to hyperglycemia-related AEs. All cases were managed safely and successfully: the deterioration of the glucose metabolism required an increase in the dose (in two patients) or the addition of new hypoglycemic drugs (in three patients) to achieve the optimal glycemic control. The preferred drugs were metformin and semaglutide. There was no need to introduce insulin in any of the patients. Detailed data on the adjustments to antidiabetic drugs are provided in Table 4.

### 3.6. Safety

Pasireotide was generally well tolerated, with the AEs mainly mild and manageable. The most common side effect was pasireotide-related hyperglycemia, which occurred in all patients analyzed, as described above. Two patients reported moderate hypocortisolism-related AEs. The symptoms of low cortisol were usually observed for several days after pasireotide injection, which required the transient use of a single morning dose of hydrocortisone. Transient gastrointestinal symptoms (abdominal pain and diarrhea) occurred in two patients. No increase in the bilirubin concentration was observed. The analysis of bilirubin levels showed relatively stable mean [SD] values over time, with a slight decrease from the baseline (0.51 mg/dL [0.09]) to the LFU (0.41 mg/dL [0.13]). However, the changes in the bilirubin levels were minimal and not statistically significant over time. Two patients experienced a rise in ALT and AST activity (exceeding 3x the ULN in one, which required the temporary discontinuation of pasireotide LAR treatment). One patient also experienced a transient mild increase in GGT activity. The analysis of ALT, AST and GGT levels showed fluctuations over time, with the mean [SD] values ranging from the baseline (29.42 IU/mL [13.26], 20.88 IU/mL [10.23], and 34.33 IU/mL [24.86], respectively) to the LFU (39.67 IU/mL [29.15], 34.17 IU/mL [32.98], and 41.67 IU/mL [34.67], respectively). The post hoc Dunn’s test comparing the baseline to the subsequent time points revealed no statistically significant differences. A complete evaluation of the liver parameters at each time point during pasireotide LAR treatment is detailed in Table 3. One patient, who had a normal gallbladder upon ultrasonographic examination at baseline, displayed detectable biliary sludge during treatment. Hair loss was reported in one patient. No significant QT prolongation was observed in any of the patients.

## 4. Discussion

This analysis reports the efficacy and safety of treatment with pasireotide LAR in patients with CD in a real-life setting based on the experience of a single center in Poland. To our knowledge, this is the first report of pasireotide LAR treatment efficacy in Poland in patients with CD.

The results of our observational study are mostly consistent with those obtained from clinical trials. In our cohort, the mean mUFC showed a significant and sustained decrease from the baseline to the LFU with statistical significance at all analyzed time points, with the most pronounced decrease in the first 3 months of therapy. mUFC normalization upon pasireotide-LAR treatment was achieved in all four patients with an elevated mUFC at baseline; however, during the entire analyzed observation period in one patient, there was an escape from pasireotide-LAR’s initial efficacy, and no further normalization of the mUFC, despite the escalation of therapy, was achieved.

Pasireotide LAR therapy was discontinued in half of the analyzed patients due to its long-term ineffectiveness, including two patients with initially normal mUFC, where the decision to stop treatment was based on persistently elevated mLNSC. Initial disease control with mUFC normalization was achieved in another patient, but an escape from pasireotide LAR efficacy was observed during further follow-ups. The analysis of mean late-night salivary cortisol showed fluctuations over time, with the largest mean reduction in mLNSC at 3 months. mUFC and mLNSC are considered to be complementary measurements for monitoring pharmacological treatment responses in CD. However, the best clinical improvement is seen in patients with the normalization of both parameters [15]. Identifying patients who could benefit from pasireotide LAR treatment is essential for efficient patient management. Research indicates that lower baseline mUFC levels are associated with higher rates of UFC normalization in response to pasireotide treatment [8,16]. However, studies evaluating the efficacy of pasireotide LAR therapy did not include patients with CD and normal mUFC. It would seem that these patients should respond very well to pasireotide LAR therapy. However, our observations show that pasireotide LAR was not able to restore this group of CD patients’ normal circadian rhythms of cortisol secretion; the clinical effects in these patients were unsatisfactory due to the persistently disturbed circadian cortisol secretion with elevated LNSC. The patients who completed pasireotide LAR therapy were switched to osilodrostat and are now well controlled. It should be noted, however, that the group of patients we analyzed who are continuing treatment with pasyreotide LAR are the patients who have a history of pituitary radiotherapy. Thus, the potential influence of baseline radiotherapy treatment on the subsequent improved response to pasireotide LAR therapy cannot be entirely excluded.

Ubiquitin-specific protease 8 (*USP-8*) gene mutational status could be a potential marker of the pasireotide response, as mutant *USP8* forms may upregulate SSTR5 transcription [1,17,18]. Indeed, recent in vitro studies report a significantly better response to pasireotide treatment in human corticotroph tumors carrying *USP8* mutations [17,19]. Studies report that *USP8*-mutant tumors are smaller, but have worse long-term postoperative outcomes compared to wild-type tumors, with a tendency toward shorter recurrence-free survival in *USP8*-mutant patients [17,20].

The analysis of ACTH levels showed relatively stable mean values over time. A detailed analysis of tumor volume was not possible in the presented group because, in three patients, the pituitary tumors were not detectable on MRI at either baseline or LFU. In the other three, the sizes of the pituitary tumors remained stable, including a patient with a large (54 × 45 × 47 mm) solid-cystic tumor after repeated TSS, Gamma Knife radiosurgery, and temozolomide treatment. One factor that should be considered in the selection of pharmacological therapy for patients with CD should also be the size of the pituitary tumor. In patients with corticotroph macroadenomas, pasireotide may decrease the size of the tumor, or at least limit its potential for further growth [21]. Pasireotide LAR therapy may also be considered in such cases as an add-on therapy to steroidogenesis inhibitors [22], especially considering the growing number of reports of corticotroph tumor progression during osilodrostat treatment [23,24].

Considering all this, pasireotide LAR therapy seems effective in a particular group of patients with CD. The most significant benefit from the treatment may be obtained by patients with a milder form of the disease (mUFC < 2x the ULN), but at the same time with a specific secretion profile (where the increased LNSC is not the dominant abnormality). Additionally, patients with macroadenomas may benefit from the stabilization/reduction in tumor size. The *USP8* mutation status may be helpful in selecting treatment.

A decrease in the mean SBP values from the baseline to the LFU was observed in the analyzed group, with the most pronounced effect after 6 months of pasireotide LAR therapy. The analysis of DBP showed a slight decrease in the mean values from baseline during the observation period. In two patients, reducing the doses of antihypertensive drugs was possible. Interestingly, a reduction in BP values was also observed in the patients in whom hypercortisolemia control was not achieved. However, according to clinical trial data, pasireotide treatment improves hypertension control independently from cortisol reduction [25]. The body weight and BMI in the analyzed group gradually decreased during the observation, but no statistical significance was found. The lipid profile assessment revealed a gradual decrease in the TC, LDL-C, HDL-C, and TGs upon analysis. Interestingly, statistical significance was found only for the HDL-C decrease at months 9 and 12.

Hyperglycemia is a known side effect of pasireotide treatment. The stimulation of pancreatic SSTR-5 suppresses insulin secretion, which, combined with negative effects on incretin hormone secretion, leads to pasireotide-induced hyperglycemia. [26,27]. Previous data also indicate that if hyperglycemia occurs, it is most likely during the first three months of treatment with pasireotide, with stabilization over time [28,29,30].

Pasireotide-related hyperglycemia occurred in all analyzed patients. Consistent with previous studies on pasireotide, the hyperglycemia-related AEs were mainly mild and manageable. However, in the analyzed group, gradually increasing FPG and HbA1c levels were observed, with the greatest differences in relation to the initial value and LOV. That difference in the time of the most significant deterioration in the glucose metabolism compared to earlier reports may be due to several factors. Firstly, the risk of hyperglycemia was closely monitored in the analyzed group, as nearly all patients, except for one, exhibited signs of impaired glucose metabolism upon entering the study. Most patients were taking glucose-lowering drugs before starting treatment with pasireotide-LAR. The initial deterioration in the glucose metabolism control in the first months of treatment led to the intensification of hypoglycemic treatment, which may not reflect the actual effects of pasireotide-LAR therapy in the first analyzed time points. Additionally, at the LFU, in half of the patients, the CD was not controlled biochemically, which was associated with an escape from pasireotide-LAR initial efficacy, and the active hypercortisolism likely worsened the glucose metabolism control. None of the patients discontinued pasireotide treatment because of hyperglycemia.

At our center, we recommend closely monitoring blood glucose in patients treated with pasireotide and management with antidiabetic medications, focusing on metformin and/or incretin-based therapy. Experts indeed recommend that after lifestyle changes and pharmacotherapy with metformin, combination therapy with agents targeting the incretin pathway is recommended [31,32,33,34]. Some experts go even further and propose that when, in pasireotide-induced hyperglycemia, pharmacotherapy is necessary, incretin-based medications should be considered as first-line treatment as an alternative to metformin [33,35].

It is also worth mentioning that the glucose metabolism status returned to the initial in the three patients in whom pasireotide was discontinued due to lack of effectiveness; that supports previous observations that pasireotide-related hyperglycemia is reversible upon treatment discontinuation [36]. To conclude, the hyperglycemia observed during pasireotide treatment is manageable in most patients without the need for treatment discontinuation. The risk factors may be used to identify patients who require more careful and proactive monitoring to optimize their outcomes during pasireotide treatment, thereby ensuring continuous treatment and improved results.

In two patients, symptoms of adrenal insufficiency were observed during pasireotide LAR therapy. Low morning serum cortisol levels were usually observed for several days after pasireotide LAR injection, which required the transient implementation of a morning dose of a hydrocortisone substitution. Interestingly, these were the two patients who had a normal mUFC at baseline. That may indicate that treatment with pasireotide LAR in patients with such a secretory profile of the pituitary tumor is not able to normalize mLNSC, on the one hand, and on the other hand, leads to low morning cortisol levels and the need for temporary hydrocortisone substitution during the day. Both patients are currently well controlled with a single evening dose of osilodrostat. Less commonly reported AEs, in line with the previous evidence, included gastrointestinal disturbances and transient elevations of liver enzyme activity. In one patient, the ALT activity exceeded 3x the ULN, but this occurred while the patient was using alternative therapies (unspecified herbs); therefore, the actual effect of pasireotide LAR cannot be fully assessed. Cholelithiasis is a recognized AE associated with long-term treatment with SRLs [26]. In our cohort, three patients were post-cholecystectomy at baseline. One patient who had a normal gallbladder upon ultrasonographic examination at baseline developed detectable biliary sludge during pasireotide LAR treatment managed with ursodeoxycholic acid. Two patients presented with gallbladder polyps at baseline, whose size did not change during observation. One patient experienced significant hair loss that occurred after more than 6 months of therapy and resolved when pasireotide LAR was discontinued.

The study’s main limitations are its small number and retrospective nature. However, considering the rarity of CD, the limited number of patients treated with pasireotide LAR in Poland, and the lack of previously published data in this area from our country, it remains representative. Another limitation is the lack of complete data on the mLNSC in two patients, which did not allow for a detailed analysis of this parameter.

## 5. Conclusions

In conclusion, the current study, based on real-world evidence and performed on a limited cohort of patients, demonstrated that treatment with pasireotide LAR is an effective therapeutic option for selected patients with CD. However, even in the patients whose complete biochemical control of the disease was not achieved, pasireotide LAR improved BP control, improved anthropometric parameters including the weight and BMI, and ameliorated the lipid profile. Glucose metabolism impairment was the most common adverse effect, and it can usually be successfully controlled. Patient-tailored, effective medical therapies are needed to optimize the long-term outcomes for patients with persistent CD.

## Figures and Tables

**Figure 1 jcm-14-02794-f001:**
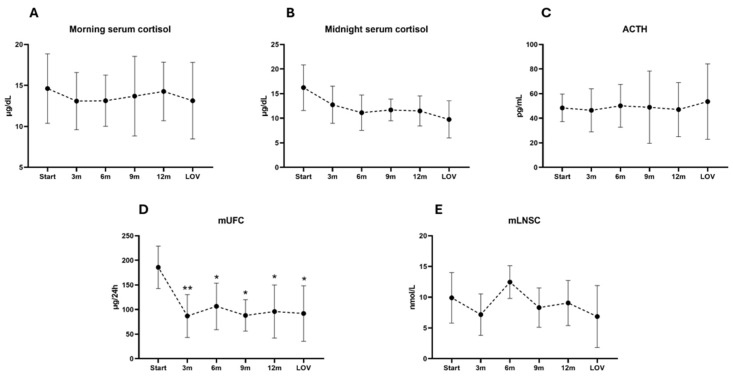
Mean and SD changes in morning serum cortisol (**A**), midnight serum cortisol (**B**), ACTH (**C**), mUFC (**D**), and mLNSC (**E**) throughout study. * is provided when significant difference (*p* < 0.05) compared to baseline was found, and ** when *p* < 0.01. Abbreviations: ACTH, adrenocorticotropin; LOV, last observed value; mLNSC, mean late-night salivary cortisol; m, months; mean late-night salivary cortisol; mUFC, mean urinary free cortisol.

**Figure 2 jcm-14-02794-f002:**
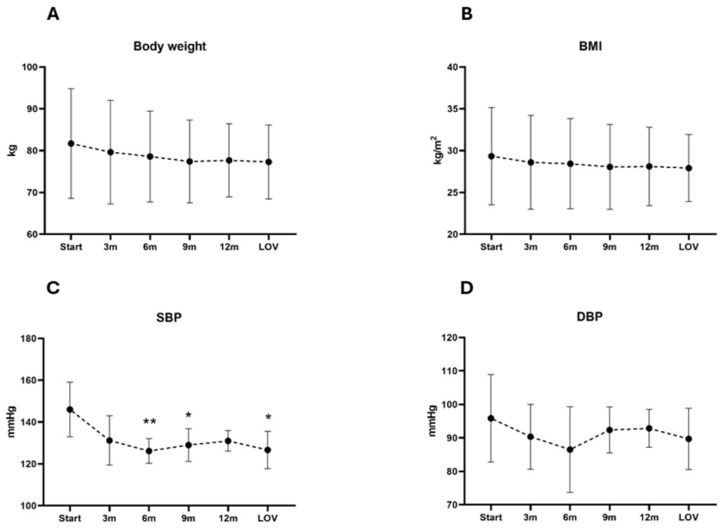
Mean and SD changes in weight (**A**), BMI (**B**), SBP (**C**), and DBP (**D**) throughout study. * is provided when significant difference (*p* < 0.05) compared to baseline was found, and ** when *p* < 0.01. Abbreviations: BMI, body mass index; DBP, diastolic blood pressure; LOV, last observed value; SBP, systolic blood pressure.

**Figure 3 jcm-14-02794-f003:**
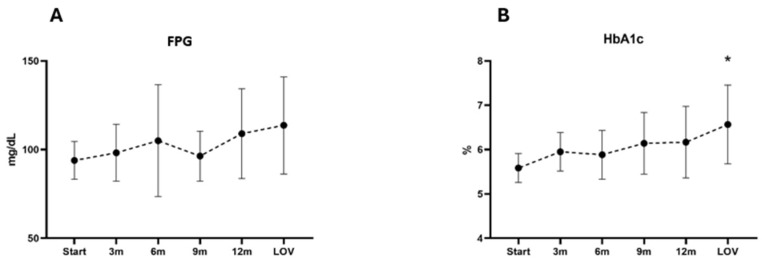
Mean and SD changes in FPG (**A**) and HbA1c (**B**) throughout study. * is provided when significant difference (*p* < 0.05) compared to baseline was found. Abbreviations: FPG, fasting plasma glucose; HbA1c, glycated hemoglobin; LOV, last observed value.

**Table 1 jcm-14-02794-t001:** Patient characteristics.

	Patient 1	Patient 2	Patient 3	Patient 4	Patient 5	Patient 6
Sex	F	F	F	M	F	F
Diagnosis	Recurrent CD	Recurrent CD	Persistent CD	Persistent CD	Recurrent CD	Persistent CD
Number of neurosurgeries	1	1	1	2	2	2
Other therapies prior to pasireotide	-	-	Osilodrostat RTH	Ketoconazole Temozolomide RTH	-	Ketoconazole RTH
Comorbidities	Hypertension IGT Dyslipidemia	Hypertension IGT Dyslipidemia	Hypertension DM2 Dyslipidemia Osteoporosis	Hypertension DM2 Dyslipidemia Osteoporosis	Hypertension IGT Dyslipidemia	Hypertension Dyslipidemia
Pituitary MRI	U					
Prior to pasireotide	m, 3 mm	Undetectable	Undetectable	M, 54 × 45 × 47 mm	m, 8 × 7 × 5 mm	Undetectable
At LFU	m, 3 mm	Undetectable	Undetectable	M, 55 × 40 × 47 mm	m, 7 × 7 × 4 mm	Undetectable
mUFC (μg/24 h)	U					
Prior to pasireotide	132.0	144.0	217.7	192.4	245.6	182.4
At LFU	92.2	113.0	55.8	36.4	192.2	59.0
mLNSC (ng/mL)	U					
Prior to pasireotide	12.8	7.0	6.9	Not available	12.9	Not available
At LFU	10.8	7.4	2.4	0.1	13.6	6.8
Max pasireotide LAR dose (mg/28 days)	20	40	20	20	40	10
Pasireotide LAR side effects	Hyperglycemia Hypercortisolism Elevated ALT and AST Hair loss	Hyperglycemia Hypocortisolism	Hyperglycemia Detectable biliary sludge	Hyperglycemia Elevated ALT, AST, and GGT Abdominal pain and diarrhea	Hyperglycemia	Hyperglycemia Abdominal pain and diarrhea
Status of pasireotide treatment	Stopped because of lack of biochemical control	Stopped because of lack of biochemical control	Ongoing	Ongoing	Stopped because of lack of biochemical control	Ongoing

Abbreviations: ALT—alanine transaminase; AST—aspartate transaminaze; CD—Cushing’s disease; DM2—diabetes mellitus type 2; GGT—gamma-glutamyltransferase; IGT—impaired glucose tolerance; mLNSC—mean late-night salivary cortisol; mUFC—mean urinary free cortisol; LFU—last follow-up; MRI—magnetic resonance imaging; RTH—radiotherapy.

**Table 2 jcm-14-02794-t002:** Hormonal parameter changes during pasireotide LAR treatment in analyzed group.

	Baseline	3 Months		6 Months		9 Months		12 Months		LOV	
	Mean (SD)	Mean (SD)	*p*-Value ^a^	Mean (SD)	*p*-Value ^a^	Mean (SD)	*p*-Value ^a^	Mean (SD)	*p*-Value ^a^	Mean (SD)	*p*-Value ^a^
ACTH, pg/mL	48.38 (11.17)	46.43 (17.57)	0.94	50.1 (17.42)	0.98	48.9 (29.39)	0.99	47.03 (21.99)	0.99	53.51 (30.68)	0.95
Morning serum cortisol, μg/dL	14.62 (4.24)	13.08 (3.49)	0.64	13.13 (3.11)	0.73	13.7 (4.86)	0.99	14.27 (3.57)	0.99	13.13 (4.67)	0.95
Midnight serum cortisol, μg/dL	16.22 (4.64)	12.73 (3.76)	0.66	11.13 (3.6)	0.38	11.68 (2.19)	0.15	11.48 (3.07)	0.11	9.75 (3.77)	0.05
mUFC, μg/24 h	185.68 (43.11)	86.72 (43.54)	0.007	106.53 (47.33)	0.02	87.99 (31.83)	0.01	95.85 (53.9)	0.03	91.93 (56.54)	0.04
mLNSC, ng/mL	9.9 (4.1)	7.17 (3.36)	0.16	12.46 (2.67)	0.88	8.32 (3.2)	0.79	9.07 (3.67)	0.82	6.85 (5.04)	0.60

**^a^** Dunnett’s post hoc test, comparison to baseline. Abbreviations: ACTH, adrenocorticotropin; LAR, long-acting release; LOV, last observed value; mLNSC, mean late-night salivary cortisol; mUFC, mean urinary free cortisol.

**Table 3 jcm-14-02794-t003:** Metabolic, cardiovascular, and liver parameter changes during pasireotide LAR treatment in analyzed group.

.	Baseline	3 Months		6 Months		9 Months		12 Months		LOV	
	Mean (SD)	Mean (SD)	*p*-Value ^a^	Mean (SD)	*p*-Value ^a^	Mean (SD)	*p*-Value ^a^	Mean (SD)	*p*-Value ^a^	Mean (SD)	*p*-Value ^a^
Body weight, kg	81.7 (13.08)	79.63 (12.39)	0.15	78.58 10.86)	0.19	77.4 (9.90)	0.2027	77.67 (8.72)	0.35	77.3 (8.83)	0.45
BMI, kg/m^2^	29.33 (5.80)	28.59 (5.60)	0.10	28.44 (5.40)	0.34	28.04 (5.08)	0.3213	28.11 (4.69)	0.47	27.91 (4.00)	0.49
SBP, mmHg	146 (13.04)	131.17 (11.79)	0.07	126.17 (5.95)	0.005	129 (7.85)	0.02	131 (4.94)	0.11	126.67 (9.00)	0.03
DBP, mmHg	95.83 (13.06)	90.33 (9.71)	0.75	86.5 (12.82)	0.45	92.33 (6.86)	0.86	92.83 (5.71)	0.94	89.67 (9.16)	0.50
FPG, mg/dL	93.9 (10.65)	98.17 (16.07)	0.53	105 (31.52)	0.63	96.33 (14.07)	0.97	109 (25.3)	0.27	113.67 (27.33)	0.12
HbA1c, %	5.59 (0.33)	5.95 (0.44)	0.14	5.88 (0.55)	0.46	6.14 (0.69)	0.25	6.17 (0.80)	0.27	6.57 (0.89)	0.04
ALT, IU/mL	29.42 (13.26)	52.83 (64.57)	0.82	35.83 (14.11)	0.30	29.63 (9.90)	0.99	25.67 (9.09)	0.88	39.67 (29.15)	0.88
AST, IU/mL	20.88 (10.23)	24.67 (18.20)	0.98	21.67 (5.79)	0.99	22.4 (4.76)	0.99	19.83 (3.13)	0.99	34.17 (32.98)	0.56
GGT, IU/mL	34.33 (24.86)	37.17 (22.69)	0.93	35.67 (21.49)	0.99	26.33 (23.69)	0.56	30.33 (13.63)	0.97	41.67 (34.76)	0.54
Bilirubin, mg/dL	0.51 (0.09)	0.53 (0.13)	0.99	0.56 (0.17)	0.90	0.48 (0.18)	0.99	0.43 (0.14)	0.64	0.41 (0.13)	0.64
TC, mg/dL	222.5 (61.48)	180 (33.89)	0.35	157 (21.06)	0.20	183.42 (74.92)	0.12	166.83 (36.75)	0.11	156.83 (38.35)	0.01
HDL-C, mg/dL	50.47 (9.67)	45 (11.45)	0.05	41.33 (15.29)	0.21	42.33 (8.07)	0.01	41.5 (9.22)	0.01	42.67 (12.71)	0.16
LDL-C, mg/dL	134.6 (53.74)	104.33 (34.93)	0.52	87 (24.02)	0.30	108.25 (62.9)	0.20	96.17 (30.20)	0.21	87.5 (33.41)	0.06
TGs, mg/dL	185.82 (62.44)	153 (65.67)	0.14	146.17 (80.13)	0.22	161.58 (88.72)	0.79	145.33 (73.57)	0.37	128.5 (45.76)	0.06

**^a^** Dunnett’s post hoc test, comparison to baseline. Abbreviations: ALT, alanine aminotransferase; AST, aspartate aminotransferase; BMI, body mass index; DBP, diastolic blood pressure; FPG, fasting plasma glucose; GGT, gamma-glutamyltransferase; HbA1c, glycated hemoglobin; HDL-C, high-density lipoprotein cholesterol; LAR, long-acting release; LDL-C, low-density lipoprotein cholesterol; LOV, last observed value; SBP, systolic blood pressure; TC, total cholesterol; TGs, triglicerides.

**Table 4 jcm-14-02794-t004:** Glycemic status and antidiabetic treatment during pasireotide LAR treatment in analyzed group.

		Baseline	6 Months	12 Months	LFU
Patient 1	Status	Prediabetes	Prediabetes	Prediabetes	Prediabetes
	HbA1c, %	5.3	5.0	5.7	5.7
	Treatment	S ^a^, 0.25 mg/week	S ^a^, 0.25 mg/week	S ^a^, 0.5 mg/week	S ^a^, 0.5 mg/week
Patient 2	Status	Prediabetes	Prediabetes	Prediabetes	Prediabetes
	HbA1c, %	5.7	5.7	5.9	5.9
	Treatment	M, 1.5 g/day	M, 1.5 g/day S ^a^, 0.25 mg/week	M, 1.5 g/day S ^a^, 0.5 mg/week	M, 1.5 g/day S ^a^, 0.5 mg/week
Patient 3	Status	Diabetes	Diabetes	Diabetes	Diabetes
	HbA1c, %	5.6	6.5	7.1	7.3
	Treatment	S ^b^, 7 mg/day	S ^b^, 7 mg/day M-XR, 1 g/day	S ^b^, 7 mg/day M-XR, 1 g/day	S ^b^, 14 mg/day M-XR, 2 g/day D, 10 mg/day
Patient 4	Status	Diabetes	Diabetes	Diabetes	Diabetes
	HbA1c, %	6.0	6.0	5.9	7.6
	Treatment	M, 1.5 g/day	M, 1.5 g/day	M, 1.5 g/day	M, 2.550 g/day
Patient 5	Status	Prediabetes	Diabetes	Diabetes	Diabetes
	HbA1c, %	5.8	6.4	7.2	7.2
	Treatment, dose	M-XR, 1 g/day	M-XR, 2 g/day	S ^a^, 0.5 mg/week M-XR, 2 g/day	S ^a^, 0.5 mg/week M-XR, 2 g/day
Patient 6	Status	Normal glycemic status	Prediabetes	Prediabetes	Prediabetes
	HbA1c, %	5.1	5.7	5.2	5.7
	Treatment	-	Diet	Diet	Diet

**^a^** Subcutaneous formulation of semaglutide. **^b^** Oral formulation of semaglutide.

## Data Availability

The data analyzed or generated during the study are available from the corresponding author on reasonable request.

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
