# Peer review of "Real-World Experience with Pasireotide-LAR in Cushing’s Disease: Single-Center 12-Month Observational Study"

_jcm, 2025, doi:10.3390/jcm14082794_

Round 1

Reviewer 1 Report

Comments and Suggestions for Authors

In the submitted manuscript, Dzialach et al present the Real-world experience with pasireotide-LAR in Cushing disease in their center. In this cohort, mean mUFC showed a significant and sustained decrease from baseline 185.68ug/24h (43.11) to 91.93 μg/24h (56.54).However Pasireotide LAR therapy was discontinued in half of the analyzed patients due to ineffectiveness. They were switched to osilodrostat and are now well controlled. The conclusion was that pasireotide LAR is an effective therapeutic option for patients with CD. This is confusing. The conclusions are not  supported by the results.

They needs re-write of some of the descriptions. They should describe the following information: the initial dose of pasireotide LAR, any changes? concomitant pituitary deficiencies(HPT HPG axis ) ?  tumor volume shrinkage? Did patients present bradycardia ,hypothyroidism, IGF reduction?

Detailed data of all 6 patients should presented,  including previous treatment, hormonal parameters  and adverse events as Table 4. The previous radiosurgery, and temozolomide treatment might affect the UFC results

Maybe they should emphasize treatment with pasireotide significantly improved, also in the absence of a complete biochemical control, anthropometric parameters including weight and BMI, and ameliorated the lipid profile reducing cardiovascular risk.

Comments on the Quality of English Language

line 25-26 'Pasireotide-LAR was overall safe and well tolerated'

use It is safe and well tolerated,

line 92-94 'liver function parameters: alanine transaminase (ALT), aspartate transaminase (AST), gamma-glutamyltransferase (GGT), and bilirubin levels as well as lipid profile were measured every 3 months.’ 

use 'the following liver function parameters were measured: ALT, AST'

line 257-258 'Detailed data regarding the modification of treatment with glucose-lowering drugs in analyzed patients are presented in Table 4.'

change :Detailed data on adjustments to antidiabetic drugs are provided in Table 4.

Author Response

We greatly appreciate your time in reviewing our manuscript. Our responses are below.

1.

„However Pasireotide LAR therapy was discontinued in half of the analyzed patients due to ineffectiveness. They were switched to osilodrostat and are now well controlled. The conclusion was that pasireotide LAR is an effective therapeutic option for patients with CD. This is confusing. The conclusions are not  supported by the results.”

In the discussion, we thus attempted to answer that two patients had their treatment discontinued due to persistently elevated mLNSC values (normal mUFC at baseline). We completed the rationale for the discontinuation of pasireotide in the third patient, who initially achieved biochemical control of the disease but subsequently observed escape from the effects of pasireotide despite increasing the dose of the drug. We have corrected the conclusion that the LAR pasireotide is effective in a selected group of patients with CD.

2.

„They needs re-write of some of the descriptions. They should describe the following information: the initial dose of pasireotide LAR, any changes? concomitant pituitary deficiencies(HPT HPG axis ) ?  tumor volume shrinkage?Did patients present bradycardia ,hypothyroidism, IGF reduction? 

„Detailed data of all 6 patients should presented,  including previous treatment, hormonal parameters  and adverse events as Table 4. The previous radiosurgery, and temozolomide treatment might affect the UFC results.”

Considering this guidance, we completely changed the concept of Table 1. We included most of the recommended parameters and descriptions for each patient separately.

3.

„Maybe they should emphasize treatment with pasireotide significantly improved, also in the absence of a complete biochemical control, anthropometric parameters including weight and BMI, and ameliorated the lipid profile reducing cardiovascular risk.”

We have supplemented the conclusions section with the indicated content as suggested.

4. Comments on the Quality of English Language

We modified the sentences according to your suggestions.

Reviewer 2 Report

Comments and Suggestions for Authors

This is a study not surprisingly showing that pasireotide works in patients with CS. Showing real life data makes sense, like showing that morning C and ACTH assessment makes no sense:). in those patients. However, the main weakness of the study is the extremely low number of participants. The reliability of the results and evidence level is low also.

I would suggest that the authors either increase the number of patients - as pasireotide was administered in frame of the National Health Service programm I assume that there are also other centers involved and a scientific cooperation is possible. Such a study even with 30 patients would be totally acceptable.

If not,  in my opinion these results could be published only as case series, if the authors are willing to reshape the manuscript.

Minor remarks: The authors write in the abstract that "The analysis of mean late night salivary cortisol showed no 17 consistent trend in the analyzed cohort" This is not consistent with the results given in Table 1.

There is no colon after "Results" in the abstract

Author Response

We greatly appreciate your time in reviewing our manuscript. Our responses are below.

1.

I would suggest that the authors either increase the number of patients - as pasireotide was administered in frame of the National Health Service programm I assume that there are also other centers involved and a scientific cooperation is possible. Such a study even with 30 patients would be totally acceptable. If not,  in my opinion these results could be published only as case series, if the authors are willing to reshape the manuscript.”

We agree that the number of patients analyzed is small. However, Cushing's disease is an ultra-rare disease and in our opinion, an analysis that includes even a few patients is valuable. Additionally, in Poland, approximately 40 patients with Cushing's disease are currently treated with pasireotide LAR, but most of them for less than a year of therapy, which additionally limits the possibility of conducting such a form of analysis at present.

2.

“The authors write in the abstract that "The analysis of mean late night salivary cortisol showed no 17 consistent trend in the analyzed cohort" This is not consistent with the results given in Table 1.”

Thank you for your comment; we have modified the conclusion drawn from the analysis of this parameter.

3.

“There is no colon after "Results" in the abstract.”

We added colon as per comment.

Round 2

Reviewer 1 Report

Comments and Suggestions for Authors

1. The author had amended the results & table 1.The presentation of the results is now crystal clear.All three patients(Patient 3 5 6) who are still on pasireotide have undergone radiotherapy before. While Patient 1 2 4 who stopped the medicine didn‘t receive radiotherapy and any other medicine like Ketoconazole. Now the conclusion was “Pasireotide-LAR is an effective alternative treatment in selected patients with CD.”  How to distinguish whether biochemical remission is attributable to radiotherapy rather than the  effect of pasireotide. Based on the current data, can it be interpreted that pasireotide has poor efficacy in Cushing's disease without radiotherapy?

You should explain the selected patients more clear, more specific

2.The presentation of the results is a bit repetitive and verbose, as table 2 and figure 1, table 3 and figure 2. The presentation of results needs to be streamlined to avoid duplication.

3. In discussion, you mentioned“ USP8 could be the potential marker of pasireotide response”. What were your patients‘ results? Did your patients 1-6 had USP8 mutation? Is that consistent with this hypothesis?

4.The format of the table1-4 needs to be modified.  There are not so many solid lines in the three-line chart. In table 4 Patient2 not Hba1c,should be HbA1c

5.It is well known that pasireotide-induced hyperglycemia, the author did not put forward a relatively novel point of view, and the corresponding paragraphs in the discussion section should be simplified

Comments on the Quality of English Language
  1. lines 165 to 167  could be amended as ‘’Post-hoc analysis revealed statistically significant differences for SBP at 6 months, 9 months, and LFU, but no significance was observed for DBP. '
  2. line 198 changes in FPG level should be changes in FPG levels

Author Response

1. 

Thank you for this comment. We have supplemented the discussion with a short section on the potential impact of radiotherapy in patients still receiving treatment with pasireotide. 

While there was a patient with an initial good clinical response to pasireotide LAR treatment, with a subsequent escape from treatment efficacy, one may wonder if the lack of radiotherapy compared to other patients influenced such a treatment outcome. However, this hypothesis cannot be entirely appropriate in the two patients with baseline normal mUFC. Here, however, we assume (as we try to discuss) that in these cases, the lack of full clinical efficacy was due to the patient's cortisol secretory profile with a predominance of abnormal mLNSCs and even an inverted diurnal cortisol secretion profile.

In the discussion we additionally added a paragraph summarizing the attempt to select patients for pasireotide therapy.

2. 

"The presentation of the results is a bit repetitive and verbose, as table 2 and figure 1, table 3 and figure 2. The presentation of results needs to be streamlined to avoid duplication."

Indeed, the way the study results are presented overlaps a bit, but in our opinion, it is complementary, and we would like it to remain in its current form.

3. Unfortunately, only two of the patients analyzed had genetic testing for USP8 mutations. In both of them, a pathogenic variant of USP8 was confirmed. There was one patient with a good response to pasireotide (Patient 3) and one with an inadequate response (Patient 1). Due to incomplete data, we did not include this information in the manuscript, as it does not allow us to formulate appropriate conclusions.

However, we do not want to leave the USP8 section out of the discussion, as it primarily seeks to answer the question of who might respond well to pasireotide LAR treatment. In addition, we have supplemented the discussion with a sentence regarding the selection of patients for pasireotide LAR treatment as a response to comment No. 1.

4. 

"The format of the table1-4 needs to be modified.  There are not so many solid lines in the three-line chart."

Thank you for this comment. In our opinion, however, the tables are pretty transparent. Their reception may be hampered in their current form due to their “raw state,” and they will probably look different after JCM edits them for the final version of the publication.

"In table 4 Patient2 not Hba1c,should be HbA1c"

We have corrected this typo. 

5. 

Thank you for your comment, we have shortened the hyperglycemia section of the discussion as suggested.

6. Comments on the Quality of English Language   We modified the sentences as suggested.  

Reviewer 2 Report

Comments and Suggestions for Authors

Thank you for your effforts.

Author Response

Thank you very much for taking the time to review our manuscript.